

# Biotin-streptavidin-guided two-step pretargeting approach using PLGA for molecular ultrasound imaging and chemotherapy for ovarian cancer

Hang Zhou[1], Jing Fu[1], Qihuan Fu[1], Yujie Feng[1], Ruixia Hong[1], Pan Li[2], Zhigang Wang[2], Xiaoling Huang[3,*] and Fang Li[1,*]

[1] Ultrasound Medicine Department, Chongqing Key Laboratory of Translational Research for Cancer Metastasis and Individualized Treatment, Chongqing University Cancer Hospital, Chongqing, Shapingba District, China

[2] Ultrasound Department, Chongqing Key Laboratory of Ultrasound Molecular Imaging, The Second Affiliated Hospital of Chongqing Medical University, Chongqing, Yuzhong District, China

[3] Department of Ultrasound, the First Affiliated Hospital of Chongqing Medical University, Chongqing, Yuzhong District, China

[*] These authors contributed equally to this work.

Corresponding authors
Xiaoling Huang,
huangxiaoling_4@163.com
Fang Li, 1768308607@qq.com

## ABSTRACT

**Background**. Ovarian cancer seriously threatens the lives and health of women, and early diagnosis and treatment are still challenging. Pre-targeting is a promising strategy to improve the treatment efficacy of ovarian cancer and the results of ultrasound imaging.

**Purpose**. To explore the effects of a pre-targeting strategy using streptavidin (SA) and paclitaxel (PTX)-loaded phase-shifting poly lactic-co-glycolic acid (PLGA) nanoparticles with perfluoro-n-pentane (PTX-PLGA-SA/PFPs) on the treatment and ultrasound imaging of ovarian cancer.

**Methods**. PTX-PLGA/PFPs were prepared with a single emulsion (O/W) solvent evaporation method and SA was attached using carbodiimide. The encapsulation efficiency of PTX and the release characteristics were assessed with high performance liquid chromatography. The phase-change characteristics of the PTX-PLGA-SA/PFPs were investigated. The anti-carcinoembryonic antigen (CEA) antibody (Ab) was covalently attached to PTX-PLGA/PFPs via carbodiimide to create PTX-PLGA-Ab/PFPs. The targeting efficiency of the nanoparticles and the viability of ovarian cancer SKOV3 cells were evaluated in each group using a microscope, flow cytometry, and cell counting kit 8 assays.

**Results**. THE PTX-PLGA-SA/PFPs were spheres with a size of $383.0 \pm 75.59$ nm. The encapsulation efficiency and loading capability of the nanoparticles for PTX were $71.56 \pm 6.51\%$ and $6.57 \pm 0.61\%$, respectively. PTX was burst-released up to 70% in 2–3 d. When irradiated at 7.5 W for 3 min, the PTX-PLGA-SA/PFPs visibly enhanced the ultrasonography images ($P < 0.05$). At temperatures of 45 °C and 60 °C the nanoparticles phase-shifted into micro-bubbles and the sizes increased. The binding efficiencies of SA and Ab to the PTX-PLGA/PFPs were $97.16 \pm 1.20\%$ and $92.74 \pm 5.75\%$, respectively. Pre-targeting resulted in a high binding efficacy and killing effect on SKOV3 cells ($P < 0.05$).

**PeerJ** ________________________________________________

**Conclusions**. The two-step pre-targeting process can significantly enhance the targeting ability of PTX-loaded PLGA nanoparticles for ovarian cancer cells and substantially improve the therapeutic efficacy. This technique provides a new method for ultrasonic imaging and precise chemotherapy for ovarian cancer.

# INTRODUCTION

Ovarian cancer ranks third in morbidity among female reproductive system tumors but first in fatality (*Goncalves, Jayson & Tarrier, 2008*; *Schmid & Oehler, 2014*). Although death in patients with ovarian cancer is associated with a variety of factors, there is no denying that early diagnosis and treatment are the keys to reducing mortality. At present, the treatment for ovarian cancer mainly consists of surgery and chemotherapy but conventional chemotherapy drugs have no tumor distribution specificity (*Banerjee & Kaye, 2013*; *Lopez, Banerjee & Kaye, 2013*). While attacking tumor cells, chemotherapy drugs can also kill normal cells, resulting in obvious toxic side effects. Consequently, methods for the accurate treatment of tumors are currently a research focus.

Targeted precision treatment for malignant tumors is very promising. Many researchers have employed effector molecules for imaging or treatment with ligands that directly bind to the receptors on tumor tissues; that is, direct targeting, to achieve targeted treatment of tumors (*Stachelek et al., 2020*; *Sun et al., 2019*; *Kato et al., 2019*). Traditional tumor targeting methods are mostly focused on direct targeting. Although the techniques are convenient to use, they are easily affected by the microenvironment in vivo. In addition, direct targeting methods may face problems during the covalent binding of ligands to effector molecules because the functional groups may be prevented from binding to the receptors, resulting in reduced efficacy due to competition (*Jung et al., 2007*; *Kennel et al., 1983*; *Koo & Kwon, 2018*). Therefore, improving the efficiency of tumor targeting is a challenge.

The concept of pre-targeting originates from radionuclide imaging research and has attracted great attention for tumor treatment and imaging (*Karacay et al., 2002*; *Kraeber-Bodere et al., 2015*; *Zhu, Jain & Baxter, 1998*). *Ding et al. (2019)* reported a pre-targeting strategy using technetium $^{99m}$-labeled dibenzocyclooctyne derivatives to achieve the imaging of glucose metabolism with a high tumor/blood ratio and tumor/muscle ratio. *Zhang et al., (2019)* used a two-step biotin-avidin pre-targeting strategy involving poly lactic-co-glycolic acid (PLGA) nanoparticles encapsulating perfluoro-n-pentane (PFP) for high intensity focused ultrasound (HIFU) ablation. However, this technology is rarely used in ultrasound molecular imaging and synergistic treatment research.

PLGA nanoparticles can pass through the endothelial space of the tumor neovascularization system and have become a hot research topic in recent years. Due to the controllability, low toxicity and side effects, high biodegradability, and good

biocompatibility, PLGA has been approved by the US Food and Drug Administration (FDA) as a drug delivery vehicle and has been used in various studies on tumor treatment and ultrasound imaging (*Kazi et al., 2020*; *Allavena et al., 2020*; *Alfaifi et al., 2020*). Carcinoembryonic antigen (CEA), a broad-spectrum tumor marker, is highly expressed in ovarian cancer, colon cancer, breast cancer, lung cancer, and other malignant tumors, and has been employed as a receptor that targets tumors (*Meller et al., 2011*). Paclitaxel (PTX) is a non-water-soluble broad-spectrum chemotherapeutic drug that needs to be dissolved in organic solvents such as absolute ethanol, which can cause allergic reactions. Therefore, it is necessary to find a method of administration that can effectively reduce or even avoid unnecessary side effects.

The objective of this study was to develop a two-step pre-targeting strategy using streptavidin (SA) and paclitaxel (PTX)-loaded phase-shifting poly lactic-co-glycolic acid (PLGA) nanoparticles with perfluoro-n-pentane (PTX-PLGA-SA/PFPs) and to explore the effects on the treatment and ultrasound imaging of ovarian cancer. Afterward, SA and PTX-loaded ultrasound molecular contrast agents with PFP (PTX-PLGA-SA/PFPs) were added to SKOV3 cells. Due to the high affinity of biotin and SA, low reaction requirements, and multi-level amplification, the PTX-PLGA-SA/PFPs could substantially adhere to SKOV3 cells, thereby facilitating targeted ultrasound molecular imaging and chemotherapy for tumors.

## MATERIALS AND METHODS

### Materials

PLGA-COOH (molecular weight 12,000) was obtained from Daigang Biological Engineering Co., LTD (Jinan, Shandong, China). PTX was purchased from Haoxuan Biological Co., LTD (Xi'an, Shanxi, China). Polyvinyl alcohol (PVA), 1-ethyl-3-(3-dimethylamino) propyl carbodiimide, hydrochloride (EDC), N-hydroxysuccinimide (NHS), Dimethyl sulfoxide (DMSO), 2-(N-morpholino) ethanesulfonic acid (MES) buffer, and fluorescein isothiocyanate (FITC)-labeled goat anti-rabbit IgG were purchased from Sigma (St. Louis, MO, USA). SA, phycoerythrin (PE)-labeled SA (PE-SA), anti-CEA antibody biotinylated anti-CEA antibody (Bio-Ab), the fluorescent dye 4′,6-diamidino-2-phenylindole (DAPI), and the fluorescent dye DiI were obtained from Boorson Biotechnology Co., LTD (Beijing, China). Phosphate-buffered saline (PBS), 4% paraformaldehyde, and cell counting kit 8 (CCK-8) kits were purchased from Baodu Biological Engineering Co., LTD (Wuhan, Hubei, China). PFP was obtained from Strem Chemicals (USA). Dichloromethane ($CH_2CL_2$) was purchased from Beibei Chemical Reagent Company (Chongqing, Sichuan, China). Sodium azide was obtained from Huanyu Biotechnology Co., LTD (Beijing, China). Fetal bovine serum (FBS), RPMI-1640 medium, and 0.25% Trypsin were purchased from Hyclone Biochemical Products Co., LTD (Wuhan, Hubei, China).

## Preparation of PTX-loaded PLGA nanoparticles and pure PLGA nanoparticles

PTX-loaded phase-change PLGA nanoparticles (PTX-PLGA/PFPs) were prepared *via* emulsification (O/W). Briefly, 50 mg and 5 mg of PTX were dissolved in two mL of dichloromethane and placed in an ice bath. Afterward, PFP and 5% PVA aqueous solution was added and the mixture was sonicated at 100 W for 6 min, followed by the addition of 20 mL of 2% (v/v) isopropanol aqueous solution. The reaction was stirred with a magnetic stirrer for 2–3 h to volatilize the dichloromethane, centrifugated and washed with water several times to obtain PTX-PLGA/PFPs, then stored at 4 °C. Pure nanoparticles without PTX (PLGA/PFPs) were prepared using the same protocol.

## Preparation of PTX-PLGA-SA/PFPs and PTX-PLGA-Ab/PFPs

The prepared PTX-PLGA/PFP nanoparticles were suspended in MES buffer (0.1 mol/L, pH 5.5). EDC and NHS were added and the mixture was incubated in a shaker for 45 min. The reaction was centrifuged and washed three times with water to remove EDC and NHS followed by resuspension in MES buffer (0.1 mol/L, pH 8.0). PTX-PLGA/PFP and SA were mixed at a mass ratio of 1:1. Ab was added to PTX-PLGA/PFP (PTX-PLGA/PFP: Ab = 20:1) and the mixture was stirred for 2 h at 4 °C, followed by washing with water several times to obtain PTX-PLGA-SA/PFPs and PTX-PLGA-Ab/PFPs. For fluorescence observation, PE-SA was used to prepare PTX-PLGA-SA-PE/PFPs in the same manner in the dark. PTX-PLGA-Ab/PFPs stained with DiI were incubated with FITC-labeled IgG for 1 h in a shaker at 4 °C and washed with water. The binding of SA and Ab to the PTX-PLGA/PFPs was confirmed using a confocal laser scanning microscope (CLSM) and flow cytometer.

## Characteristic of nanoparticles

The morphology of PTX-PLGA-SA/PFPs was observed with a transmission electron microscope (TEM, Hitachi, Tokyo, Japan). The particle size and zeta potential of the PTX-PLGA/PFPs, PLGA/PFPs, PTX-PLGA-Ab/PFPs, and PTX-PLGA-SA/PFPs were determined with a Malvern particle size analyzer (Brookhaven Instruments Co., Holtsville, USA). High-performance liquid chromatography (HPLC, Waters E2695, Germany) was used to detect the encapsulation efficiency and drug loading of PTX in the PTX-PLGA/PFPs. The liquid-phase conditions were as follows. The mobile phase was acetonitrile water at a ratio of 45:55, the detection wavelength was 227 nm, and the detector was the UV2489 C18 Xbridge (150 × 4.6 mm, 3.5 μm). The volume of the organic phase was two mL, the temperature was 4 °C, and the stirring speed was 800 rpm. The power capacity of the sonicator was 100 W in pulse mode, which was applied for 5 s every 5 s, with a total sonicating time of 6 min. PTX was dissolved in methanol to prepare a standard solution at concentrations of 12.5, 25, 50, 75, and 100 μg/mL and used to create a standard curve. The PTX-PLGA/PFPs were demulsified using methyl chloride and the PTX content was determined using HPLC. PTX encapsulation rate (%) = (Detection amount/total dose) × 100%. Drug loading capacity (%) = (Detection amount/total mass of nanoparticles) × 100%.

### In vitro release characteristics of PTX-loaded nanoparticles

Two mL of PTX-PLGA/PFPs nanoparticles (resuspended in PBS) was added to a dialysis bag and placed in 150 mL of medium containing 30% ethanol, 0.01% TWEEN-80, and 0.02% azide sodium. Subsequently, the mixture was placed on a shaker at 37 °C and shaken at 150 rpm. Then, one mL of the solution was removed at 2, 4, 8, 12, 24, 48, 72, 96, 120, 144, and 168 h, respectively, and one mL of fresh solution was added. The PTX content was determined using HPLC.

### Low-intensity focused ultrasound (LIFU)-induced PTX-PLGA-SA/PFPs phase change for ultrasound imaging in vitro and thermally induced phase change

PTX-PLGA-SA/PFPs nanoparticles were placed in the wells of a gel model and irradiated with LIFU at different power levels (3.3, 4.4, 5.5, 6.5, 7.5, 8.5 W) for 3 min before being observed in normal mode (B-mode) and contrast-enhanced ultrasound (CEUS). The average echo intensities of the ultrasound images before and after the phase change were compared using the DFY image analysis system. After determining the optimal power, LIFU irradiation was performed at 0.1, 0.5, 1.0, 2.0, and 5.0 mg/mL of the nanoparticle concentration (based on the quality of PLGA). In addition, the heating plate method was used to observe the thermally induced phase transition process for the nanoparticles under a microscope.

### Cell culture

Human ovarian cancer SKOV3 cells were cultured with RPMI 1640 medium containing 10% FBS and 1% penicillin-streptomycin with 5% $CO_2$ at 37 °C. The cells were cultured in a constant temperature incubator (the incubation processes described later were performed under this condition). The cells were digested with 0.25% trypsin and centrifuged at 1,000 rpm for 5 min.

### Cellular immunofluorescence experiment

SKOV3 cells in the logarithmic growth phase were cultured in a disposable 15 mm glass-bottom dish, incubated with Ab for 2 h, and washed with PBS, followed by adding FITC-labeled IgG (secondary antibody) for 1 h. The cells were fixed with 4% paraformaldehyde for 15–20 min, washed with PBS three to five times, incubated with DAPI for 5–8 min, washed with PBS three to five times, and observed under a CLSM.

### Cytotoxicity of PTX-PLGA/PFPs

SKOV3 cells were seeded in a 96-well plate at a concentration of $10^4$ cells/well and incubated with PTX-PLGA/PFPs at concentrations of 1.0, 2.0, 3.0, 4.0, and 5.0 mg/mL (calculated based on PLGA). Cells without any treatment were used as the negative controls and each group had five replicates. After 24 h or 48 h, the cells were washed with PBS three to five times, and the cell survival rate was assessed with the CCK-8 assay according to the instructions for the kit. Cell survival rate (v%) = (optical density (OD) of experimental group − OD of zero adjustment group)/(OD of negative group − OD of zero adjustment group) * 100%.

## In vitro targeting experiment

Cells were cultured in a disposable dish and divided into a drug-loaded pre-targeting group, drug-loaded direct-targeting group, drug-loaded non-targeting group, and antibody-blocking group. In the pre-targeting group, the cells were first incubated with 200 μL of Bio-Ab at a concentration of 100 μg/mL for 2 h, washed with PBS three to five times, and then incubated with one mL of DiI-labeled PTX-PLGA-SA/PFPs. In the drug-loaded direct-targeting group, nanoparticles and the same amount of DiI-PTX-PLGA-Ab/PFPs were added. In the drug-loaded non-targeting group, DiI-PTX-PLGA/PFPs were added. The procedure for the antibody blocking group was the same as that for the drug-loaded pre-targeting group, but Bio-Ab was replaced with the same amount of Ab. All cells were further incubated for 45 min, washed with PBS several times, fixed with 4% paraformaldehyde for 20 min, washed with PBS three to five times, stained with DAPI, washed with PBS, and observed under a CLSM. In addition, the cells were cultured in a six-well plate with approximately $10^6$ cells per well. The blank control group consisted of cells without any treatment. After adding the corresponding nanoparticles or reagents to each group, the cells were cultured for 45 min and washed with PBS. The cells were digested with 0.25% trypsin, centrifuged at 800 rpm, resuspended in one mL of PBS, and visualized with flow cytometry.

## In vitro toxicity to SKOV3 cells

SKOV3 cells were seeded in 96-well plates at a concentration of $10^4$ cells/well and divided into a drug-loaded pre-targeting group, drug-loaded direct-targeting group, drug-loaded non-targeted group, antibody-blocking group, nanoparticle only group, free drug group, and blank control group. Each group had five replicates. In the drug-loaded pre-targeting group, the cells were first incubated with 25 uL of Bio-Ab at a concentration of 100 ug/mL. After incubating for 2 h, the cells were washed with PBS three to five times and incubated with 100 uL PTX-PLGA-SA/PFPs. The procedure for the antibody blocking group was the same as that for the drug-loaded pre-targeting group, but Bio-Ab was replaced with the same amount of Ab. In the drug-loaded direct targeting group, PTX-PLGA-Ab/PFPs were added. In the drug-loaded non-targeting group, PTX-PLGA/PFPs were added. In the nanoparticle only group, the same amount of PLGA/PFPs was added to each well. In the free drug group, an equal amount of fresh 1640 medium containing PTX (10 ug/mL) was added to each well. In the blank control group, 100 uL of fresh 1640 medium was added to each well. After adding the corresponding nanoparticles or reagents, the cells were cultured for 45 min, washed with PBS three to five times, and 100 uL of fresh RMPI 1640 medium was added. After 24 h or 48 h of incubation, the CCK-8 assay was used to determine the cell viability.

## Statistical analysis

All statistical analysis was performed using SPSS 19.0 statistical analysis software. The independent sample t test was used for comparison between two groups and single factor analysis of variance was used to compare the means of multiple groups. The least significant difference (LSD) test was used for pairwise comparisons between multiple groups and $P < 0.05$ was considered as statistically significant.

Peer

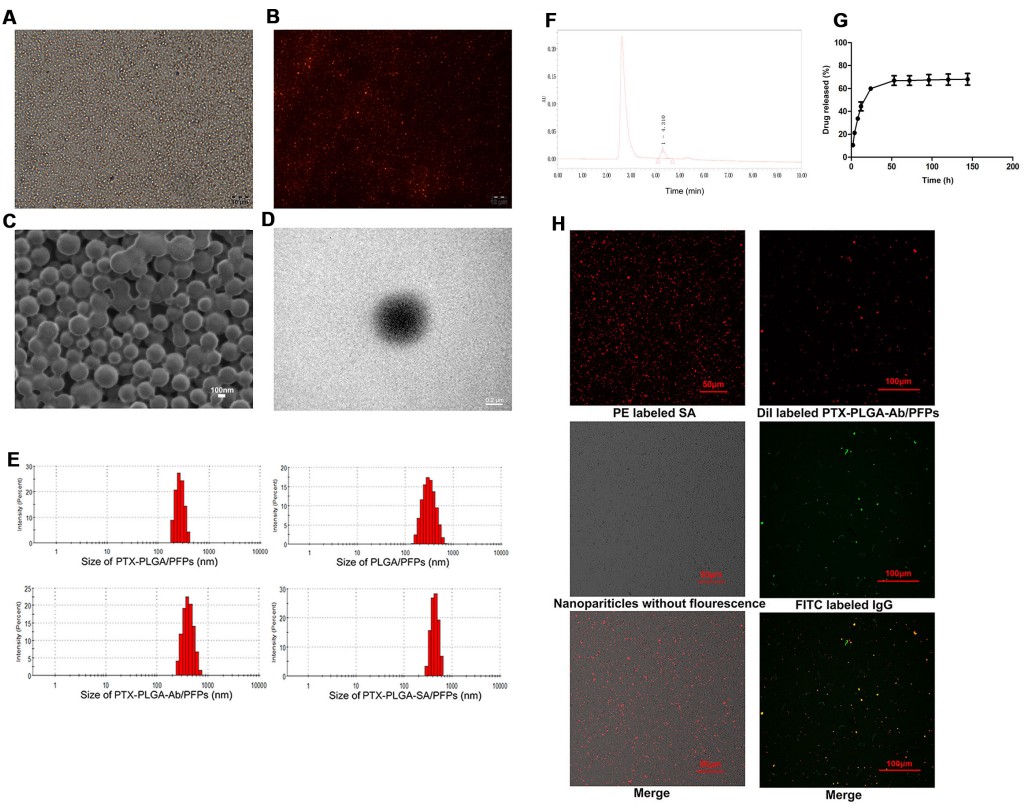

**Figure 1  Light microscope and electron microscope images of PTX-PLGA-SA/PFPs.** (A) Light microscope image. (B) Inverted fluorescence microscope image of PTX-PLGA-SA/PFPs labeled with DiI showing as red fluorescence. (C) SEM image. (D) TEM image. (E) Size distributions of PTXPLGA/PFPs, PLGA/PFPs, PTX-PLGA-Ab/PFPs, and PTX-PLGA-SA/PFPs. (F) The chromatography results for PTX encapsulated in nanoparticles in HPLC. (G) The release curve for PTX from PTXPLGA/PFPs in vitro. (H) Confocal laser scanning microscope images of PE labeled SA, DiI labeled PTX-PLGA-Ab/PFPs, Nanoparticles without fluorescence, and FITC labeled IgG.

## Result
### Preparation and characterization of PLGA

The prepared PTX-PLGA-SA/PFPs were spheres with a uniform size (Figs. 1A–1D). The diameters of the PTX-PLGA/PFPs, PLGA/PFPs, PTX-PLGA-Ab/PFPs, and PTX-PLGA-SA/PFPs were $333.0 \pm 52.27$, $364.9 \pm 106.9$, $397.7 \pm 99.96$, and $383.0 \pm 75.59$ nm, respectively (Fig. 1E). The zeta potentials of the PTX-PLGA/PFPs, PLGA/PFPs, PTX-PLGA-Ab/PFPs, and PTX-PLGA-SA/PFPs were $-15.70 \pm 3.07$, $-32.7 \pm 6.81$, $-0.93 \pm 3.43$, and $-5.66 \pm 3.46$ mV, respectively. The zeta potentials of the PTX-PLGA/PFPs without SA or Ab and PLGA/PFPs without drugs were significantly higher than those for the other nanoparticles ($P < 0.05$).

### PTX load and release

The PTX encapsulation rate and loading capacity were $71.56 \pm 6.51\%$ and $6.57 \pm 0.61\%$, respectively (Fig. 1F). PTX was released relatively quickly in the first 48 h and the drug release rate exceeded 70% after nearly 50 h. After this period, the drug release rate became

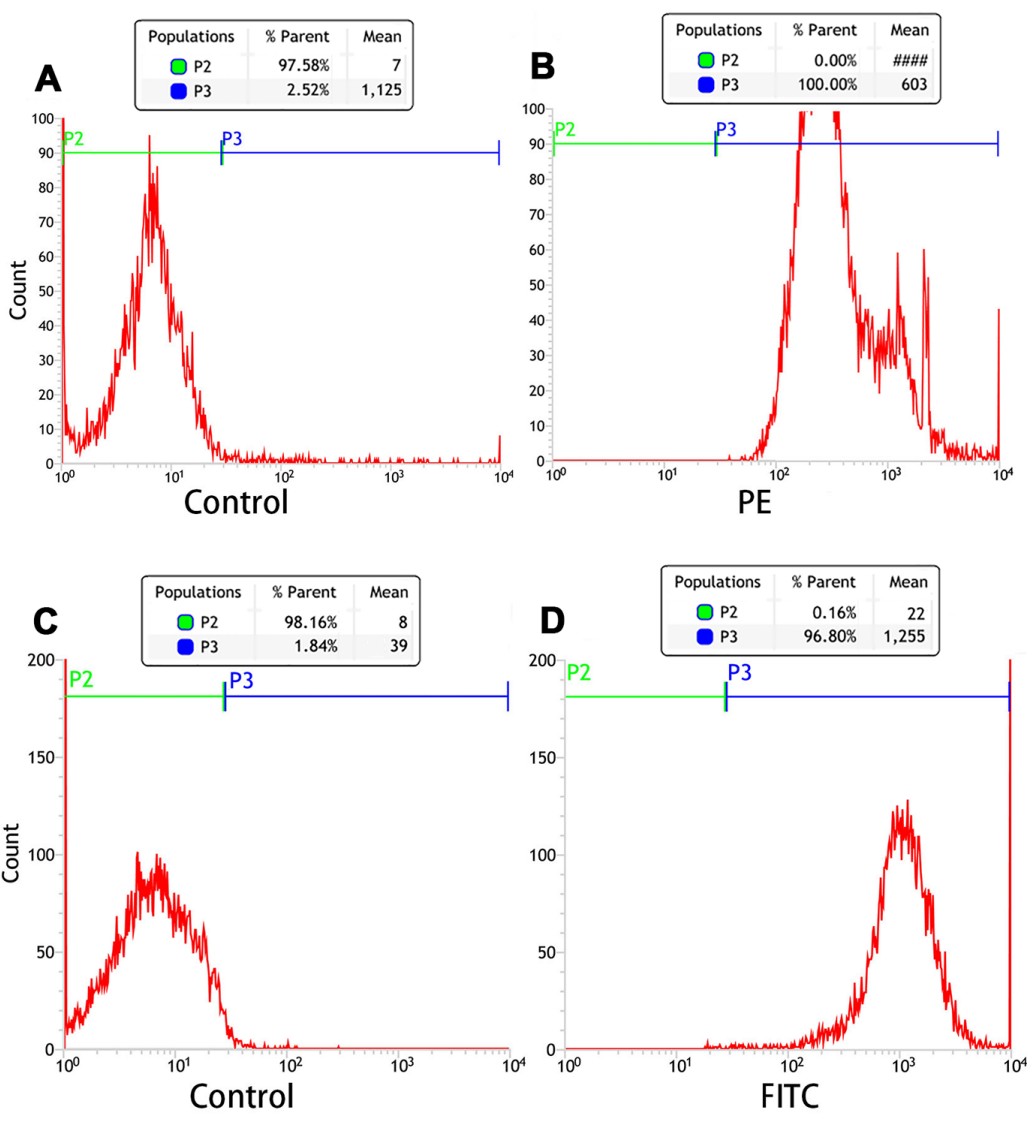

**Figure 2** Flow cytometry analysis of the connection efficiency of both SA and Ab bound to PTX-PLGA/PFPs. (A) Control; (B) SA group; (C) Control; (D) Ab group.

significantly slower, showing obvious burst-release characteristics with a high drug release rate (Fig. 1G).

## Connection between SA and Ab and PTX-PLGA/PFPs

Orange fluorescence was observed when the PE-SA attached to the fluorescent PTX-PLGA/PFPs, indicating that SA successfully attached to the nanoparticles. The fluorescence was yellow when FITC-labeled IgG attached to the DiI-labeled PTX-PLGA/PFPs, demonstrating the Ab successfully attached to the PTX-PLGA/PFPs (Fig. 1H). The mean connection rate for SA to PTX-PLGA/PFPs measured with flow cytometry was 97.16 ± 1.20%, and that of Ab was 92.74 ± 5.75% (Figs. 2A–2D).

### In vitro LIFU-induced PTX-PLGA/PFPs phase transition for ultrasound imaging

After 3 min of LIFU with ultrasound irradiation with a power of 3.3, 4.4, 5.5, 6.5, 7.5, and 8.5 W, each group exhibited significantly higher echo intensity in B-mode, and the echo intensity was highest at 6.5 W and 7.5 W ($P < 0.05$). However, the difference between the echo intensity at 6.5 W and 7.5 W was not statistically significant ($P > 0.05$). In CE-mode, the echo intensity increased with the increase of ultrasound power, reaching the highest value at 7.5 W ($P < 0.01$) but decreased at 8.5 W (Figs. 3A–3C). The echo intensity for different concentrations of nanoparticles (0.1, 0.5, 1.0, 2.0, 5.0 mg/mL) irradiated with ultrasound at 7.5 W for 3 min was investigated. The results showed that the phase change caused by LIFU had a certain relationship with the concentration. When the concentration exceeded 1.0 mg/mL, more obvious images could be obtained in B-mode, and when the concentration exceeded 2.0 mg/mL, the resolution of the CE-mode was significantly enhanced (Figs. 3D, 3E). There was no significant difference in the echo intensity between 2.0 and 5.0 mg/mL in the B-mode and CE-mode ($P > 0.05$).

### Heat-induced PTX-PLGA-SA/PFPs phase transition

The particle size of PTX-PLGA-SA/PFPs before heating was at the nanometer level and the distribution was uniform. When the temperature of the heating plate reached 45 °C, some of the nanoparticles began to undergo a phase change and gradually increased in size from nanometers to micrometers. When the temperature of the heating plate reached 60 °C, most of the nanoparticles underwent a phase change, with different particle sizes ranging from a few microns to tens of microns. Some of the microbubbles ruptured after increasing in size to a certain extent (Fig. 3F).

### Cellular immunofluorescence and viability of cells incubated with PTX-PLGA/PFPs

The products from the two-step biotin-streptavidin pre-targeting technology were combined with PTX-loaded phase-change PLGA multifunctional nanoparticles to treat ovarian cancer (Fig. 4A). Green fluorescence was observed, indicating that the FITC-labeled IgG adhered to the cell membrane and the SKOV3 cells expressed a large amount of CEA antigens (Fig. 4B). CCK-8 assay results showed that the PTX-PLGA/PFPs killed the SKOV3 cells in a concentration-dependent manner. As the concentration of the nanoparticles increased, the survival rate of the SKOV3 cells at 24 h and 48 h gradually decreased (Fig. 5A).

### Cell targeting and killing effect on tumor cells for nanoparticles in vitro

Stronger red fluorescent could be observed with CLSM, demonstrating that more DiI-labeled nanoparticles adhered to the SKOV3 cells compared with the drug-loaded direct-targeting group and the other groups (Fig. 5B). Flow cytometry results showed that the fluorescence intensity of the drug-loaded pre-targeting group was significantly stronger than that for the other groups, and the difference was statistically significant ($P < 0.05$) (Figs. 5C–5F). The fluorescence intensity of the drug-loaded direct-targeting group was higher than that for the drug-loaded non-targeting group ($P < 0.05$).

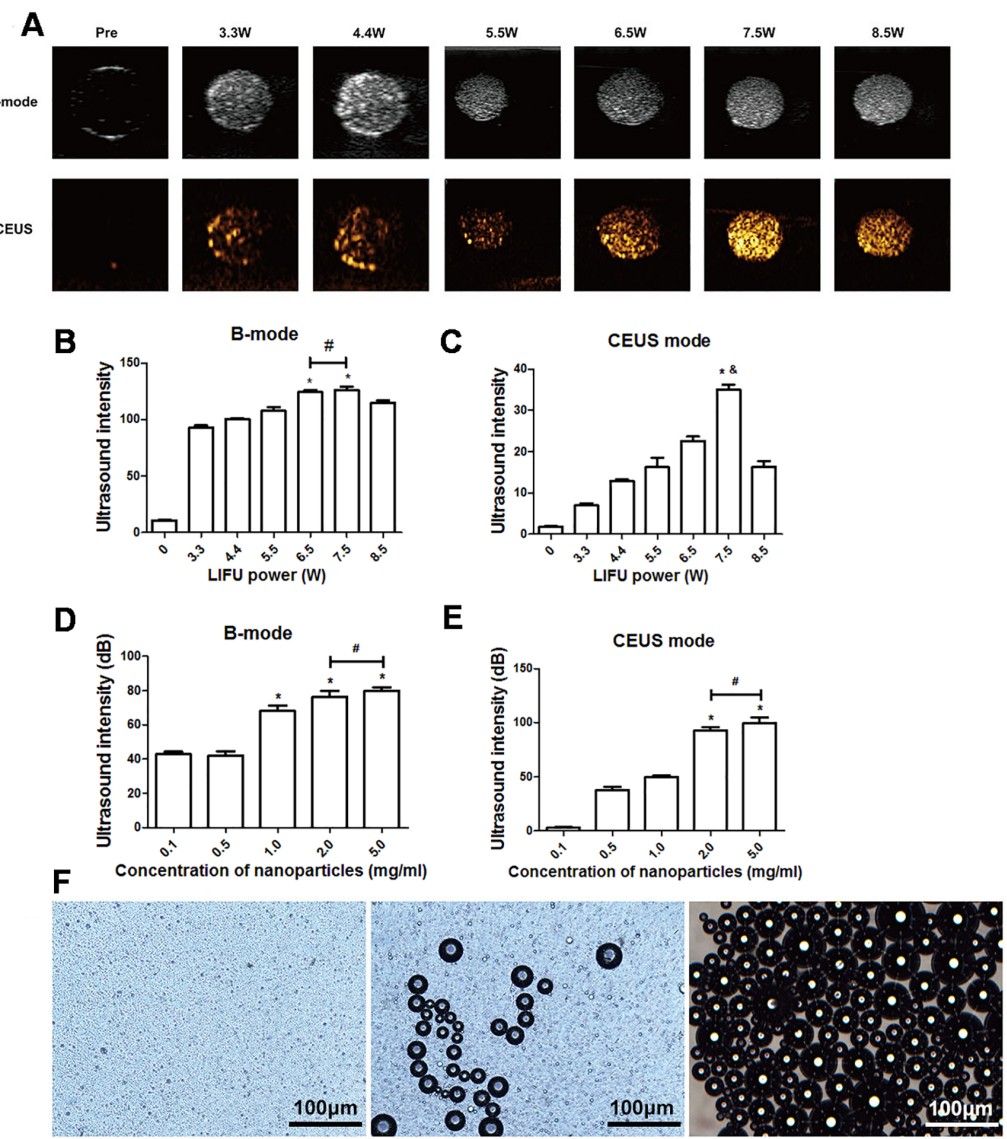

**Figure 3  Phase shifting of PTX-PLGA-SA/PFPs.** (A) After exposure to LIFU at different power levels, the B-mode and contrast ultrasonic images of nanoparticles at 3.3, 4.4, 5.5, 6.5, 7.5, and 8.5 W are shown. After irradiation with LIFU at different power levels, the average echo intensities of the nanoparticles are shown. (B) B-mode; (C) CEUS mode. An asterisk (*) indicates there is a significant difference compared with the 0.1 mg/ml group ($P < 0.05$); a number sign (#) indicates there is no significant difference between the two groups ($P > 0.05$). An ampersand (&) indicates there is a significant difference compared with the rest of the groups ($P < 0.05$). The average ultrasonic echo intensity (dB) of the nanoparticles after phase transformation induced by LIFU at different concentrations. (D) B-mode; (E) CEUS. "*" indicates there is a significant difference compared with the 0.1 mg/ml group ($P < 0.05$); "#" indicates there is no significant difference between the two groups ($P > 0.05$). (F) Light microscopic images of nanoparticles before and after phase transformation caused by heating.

The results of the CCK-8 assay showed that when incubated for 24 h, the cell survival rate for the free drug group was the lowest ($P < 0.05$) and the difference was statistically significant. The cell survival rate for the drug-loaded pre-targeting group was close to 40%,
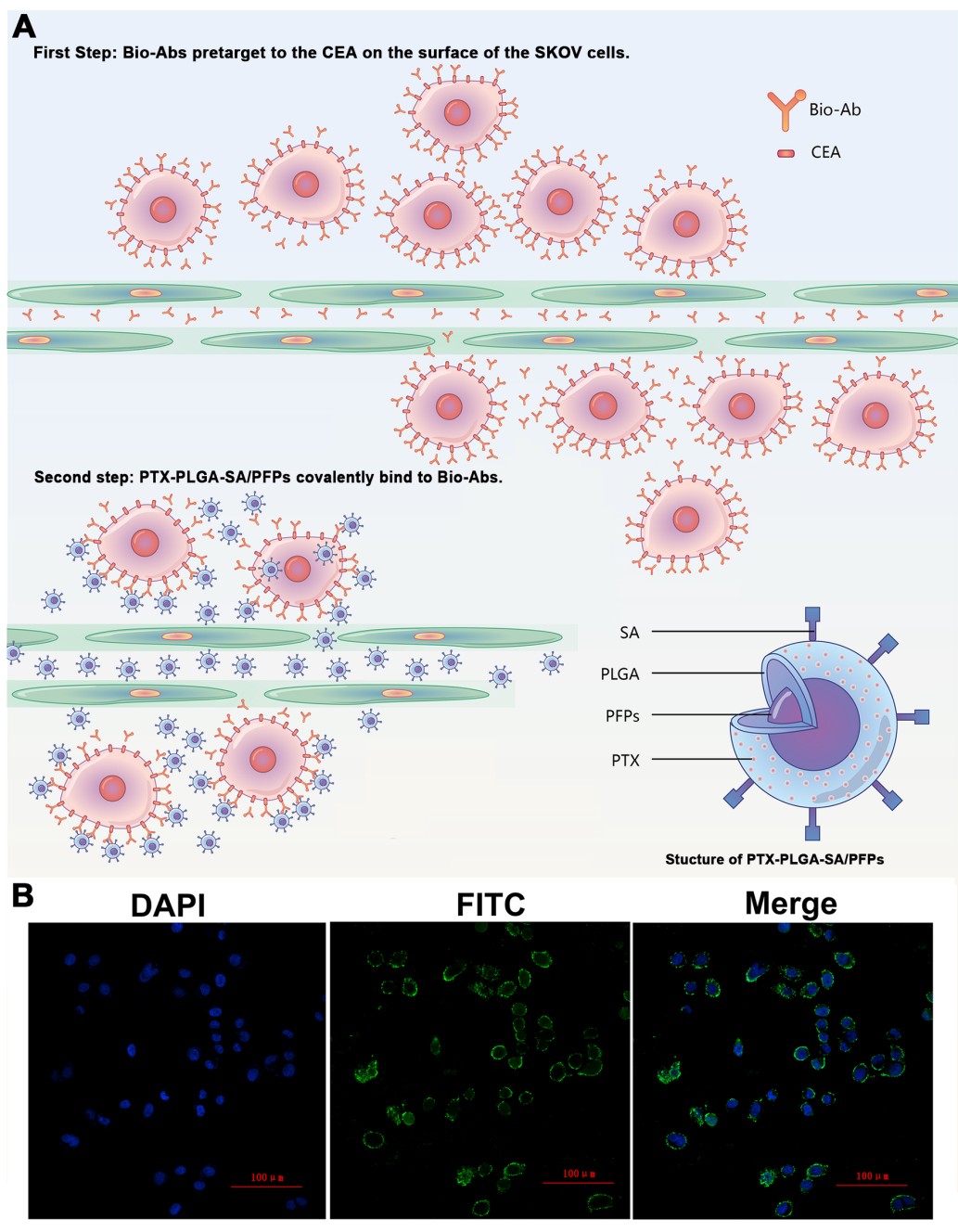

**Figure 4** **The schematic diagram of the two-step pre-targeting technology.** (A) The schematic diagram of the two-step pre-targeting technology. (B) Confocal laser scanning microscope images of human ovarian cancer SKOV3 cells showing the results of incubation with FITC-labeled secondary antibodies. Green, FITC. Blue, DAPI.

which was lower than that of the drug-loaded direct-targeting group and the other groups except for the free drug group. The difference was statistically significant ($P < 0.05$). At 48 h, the cell survival rate for the drug-loaded pre-targeting group decreased to about

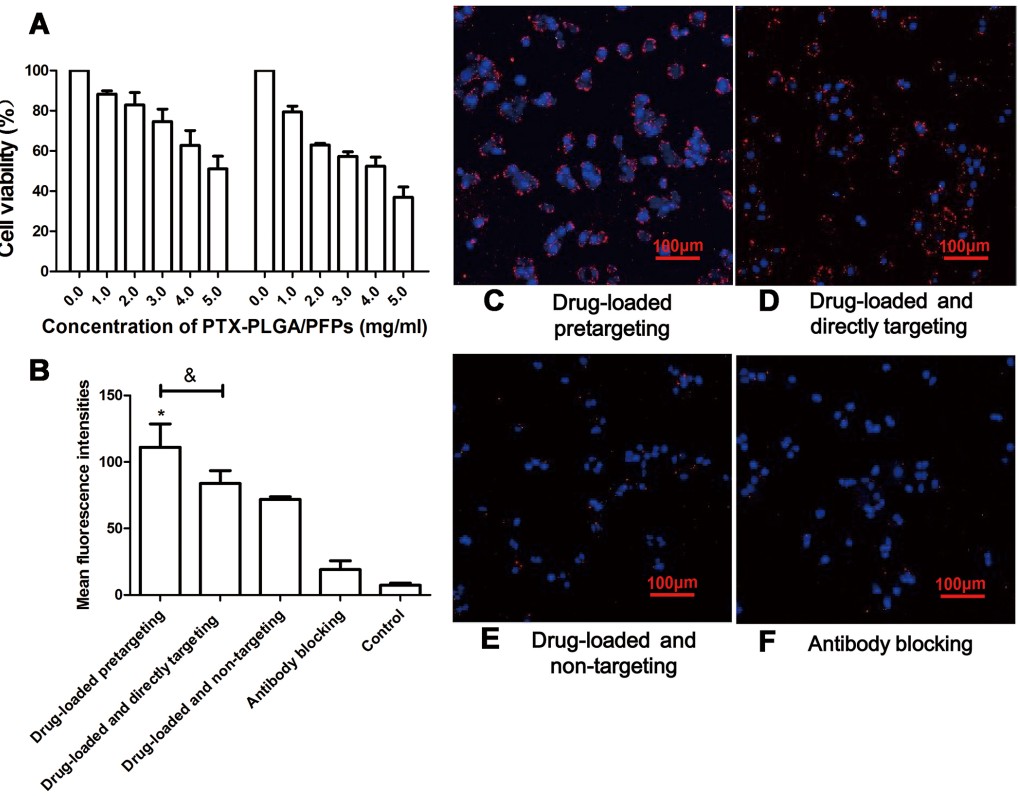

**Figure 5** **The survival rate of SKOV3 cells after adding nanoparticles and targeting efficacy of nanoparticles.** (A) Effects of different concentrations of PTX-PLGA/PFPs on the survival rate of SKOV3 cells determined via CCK-8 assays. (B) The average cell fluorescence intensities detected with flow cytometry. An asterisk (*) indicates there is a significant difference compared with the other groups ($P < 0.05$); an ampersand (&) indicates there is a significant difference compared with these two groups ($P < 0.05$). Laser confocal microscope images of SKOV3 cells incubated with (C) Drug-loaded pretargeting, (D) Drug-loaded and directly targeting, (E) Drug-loaded and non-targeting, and (F) Antibody blocking.

5%, which was lower than that for the other groups except the free drug group, and the difference was statistically significant ($P < 0.05$). At both 24 h and 48 h, the cell survival rate for the free drug group was the lowest ($P < 0.05$) (Fig. 6).

## DISCUSSION

The morbidity and fatality rates of ovarian cancer in women are very high and early diagnosis and treatment for the disease are challenging. In this study, SA and PTX-loaded phase-shifting PLGA nanoparticles (PTX-PLGA-SA/PFPs) were successfully fabricated with high encapsulation efficiency for PTX and conjugating efficiency for SA. With a burst release in vitro, PTX was released from the nanoparticles at a high rate. Phase transition was induced with LIFU and heating, and after phase transformation, the nanoparticles visibly increased the quality of contrast-enhanced ultrasonic imaging. At present, tumor-targeting PLGA nanoparticles loaded with drugs, genes, etc are a research hotspot in

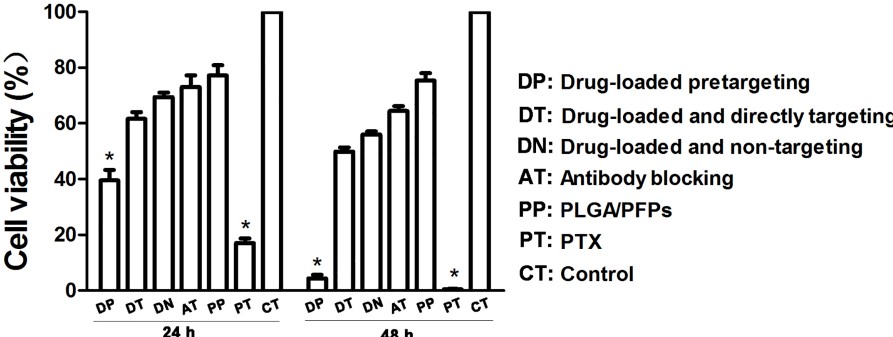

**Figure 6  Cancer cell survival rates determined with CCK-8 assays.** An asterisk (*) indicates there is a significant difference compared with the rest of the groups ($P < 0.05$).

ultrasonic molecular imaging and therapy. Loading the chemotherapeutic drug PTX into phase-shifting PLGA nanoparticles and targeting ovarian cancer cells can not only improve ultrasonic molecular imaging for the early diagnosis of ovarian cancer, but can also alleviate the drug delivery limitations caused by hydrophobicity, increase drug effectiveness, and reduce the side effects.

Direct targeting is one of the most commonly used traditional methods for tumor targeting due to the simplicity of operation but is susceptible to the microenvironment *in vivo*. In addition, using antibodies as an example, the randomness of covalent binding between the ligand and effector might cause the partial competitive inhibition of binding between functional groups and receptors, leading to a reduction in the targeting ability. Therefore, methods to improve the efficiency of tumor cell targeting have become a crucial problem which scholars are trying to solve. The antibody molecules used as ligands, regardless of their Fab or Fc segments, are compatible with PLGA. However, the connecting positions are random, which may inhibit the binding of the Fab segment to the receptor due to competition, resulting in reduced targeting.

Pre-targeting technology can effectively improve the tumor target/non-target (T/NT) ratio. This study was conducted to develop PTX-loaded and phase-shifting PLGA nanoparticles. Using a two-step biotin-SA pre-targeting technology for targeting ovarian cancer cells in vitro, we aimed to explore whether the pre-targeting technique is a feasible strategy for tumor targeting to improve ultrasonic molecular imaging and treatment. The technique provides a new approach for the accurate diagnosis and treatment of ovarian cancer. The two-step biotin-avidin pre-targeting technology has the characteristics of strong binding in vivo and multi-level amplification and has been applied in other imaging fields (*Qiu et al., 2015*). The T/NT value using a pre-targeting technique is significantly higher than that of direct targeting methods (*Hapuarachchige & Artemov, 2020*; *Hapuarachchige et al., 2020*; *Imberti et al., 2020*). However, this method is rarely used in ultrasound molecular targeting research. Avidin was replaced with SA, which is a derivative of avidin with a lower isoelectric point, no glycosyl base, lower non-specific binding affinity, and higher biosafety (*Nguyen, Sly & Conboy, 2012*; *Paganelli, Magnani & Fazio, 1993*; *Wilbur et al.,*

*1996*). In the current research, cell fluorescence immunoassays showed that a large amount of FITC-labeled secondary antibody adhered to the cell membrane area of SKOV3 cells, confirming that CEA can be expressed in large quantities in the SKOV3 cell membrane.

In this study, PTX-PLGA/PFPs were prepared using the single emulsification solvent evaporation method (O/W), which has simple operating steps and a high encapsulation rate for fat-soluble drugs. The results showed that the nanoparticles prepared with this method have a high encapsulation rate for PTX, which is consistent with related literature reports (*Sadat Tabatabaei Mirakabad et al., 2014*; *Klose et al., 2008*; *Semete et al., 2010*). SA was covalently bonded to the surface of the nanoparticles; therefore, we anticipate that it will not affect the encapsulation efficiency for PTX. For the in vitro release of PTX, neutral PBS was used as the dialysis environment, and the slow-release medium was configured according to the fat solubility characteristics of PTX. The in vitro release curve for PTX showed that the drug was burst-released and reached peak release within 2 to 3 days. It can be speculated that the release rate for PTX will be faster in vivo due to the metabolism of PLGA and the released amount would be higher. Simple nanoparticles (PLGA/PFPs) without PTX were also prepared as the control for further cytotoxicity experiments.

Direct-target PLGA nanoparticles attached to anti-CEA antibodies were prepared and the targeting efficiency was compared between the pre-targeting technique and direct targeting methods. The connection between SA and Ab and the nanoparticles was established using the carbodiimide method, which is simple to perform, has high binding efficiency, and forms a stable connection between nanoparticles and ligands. Flow cytometry was used to semi-quantitatively evaluate the connection between SA, Ab, and the PTX-PLGA/PFPs and the results showed that the connection rate for the Ab group was lower than that for the SA group. This may be because SA was labeled with PE directly, but the immunofluorescence method was indirectly used to analyze the binding of Ab through FITC-labeled secondary antibodies. In order to reduce the difficulty of LIFU-induced phase change in nanoparticles, we selected PFP, which has a boiling point of 29 °C, as the phase change material. The experimental results showed that the temperature at which the nanoparticles underwent phase change was significantly higher than that of PFP, which is consistent with the results in other studies (*Mountford, Thomas & Borden, 2015*). We speculated that this may be due to the obstacle presented by the solid shell to the vaporization and diffusion of the inner core and the influence of the Laplace force inside the nanoparticles. However, these factors are also conducive to the stability of the nanoparticles in vivo.

In the in vitro targeting experiment, the pre-targeting technique improved the targeted adhesion ability of the PTX-loaded phase-change PLGA nanoparticles to ovarian cancer SKOV3 cells. In the current study, the target nanoparticles were incubated with the cells for 45 min, which is shorter than the >2-h duration used in other studies (*Costanzo et al., 2016*; *Nsanzamahoro et al., 2020*; *Luo, Gong & Ma, 2020*). Shortening the co-incubation time for the cells and nanoparticles may reflect the time line of the connection between biotin and SA.

There are several limitations in this study. First, the optimization of the in vitro LIFU-induced phase transition conditions for the nanoparticles was insufficient and the influence of SA on the PTX drug loading was not studied. Second, the encapsulation

efficiency of liquid fluorocarbon (PFP) was not measured and the release profiles were only investigated in vitro. Third, only the cell survival rate was evaluated and the influence of the nanoparticles on the cell cycle was not studied.

In conclusion, PTX-containing phase-change PLGA nanoparticles targeting ovarian cancer SKOV3 cells were developed using a biotin-SA two-step pre-targeting process. The research results showed that the pre-targeting technology can improve the in vitro targeted adhesion ability of PTX phase-change PLGA ultrasound molecular contrast agents to ovarian cancer SKOV3 cells. Compared with direct targeting, this technique exhibited better targeting and killing of tumor cells. The results provide a good experimental foundation for further research on pre-positioned ultrasound molecular imaging technology in vivo and also provide a new idea for targeted ultrasound molecular imaging and the treatment of ovarian cancer.

### Funding

This work was supported by the Natural Science Foundation of Chongqing, China (No. cstc2020jcyj-msxmX0538), and the National Cancer Center Climbing Fund, China (No. NCC201822B75), and the Chongqing Technology Innovation and Application Development Project, China (No. cstc2019jscx-msxmX0099). The funders had no role in study design, data collection and analysis, decision to publish, or preparation of the manuscript.

### Grant Disclosures

The following grant information was disclosed by the authors:
Natural Science Foundation of Chongqing, China: cstc2020jcyj-msxmX0538.
National Cancer Center Climbing Fund, China: NCC201822B75.
Chongqing Technology Innovation and Application Development Project, China: cstc2019jscx-msxmX0099.

### Competing Interests

The authors declare there are no competing interests.

### Author Contributions

- Hang Zhou conceived and designed the experiments, performed the experiments, analyzed the data, authored or reviewed drafts of the paper, and approved the final draft.
- Jing Fu, Qihuan Fu, Yujie Feng and Ruixia Hong analyzed the data, prepared figures and/or tables, and approved the final draft.
- Pan Li and Xiaoling Huang conceived and designed the experiments, performed the experiments, authored or reviewed drafts of the paper, and approved the final draft.
- Zhigang Wang conceived and designed the experiments, analyzed the data, authored or reviewed drafts of the paper, and approved the final draft.
- Fang Li analyzed the data, authored or reviewed drafts of the paper, and approved the final draft.

## Data Availability

The raw data are all available in the Supplemental File.

## Supplemental Information

Supplemental information for this article can be found online at http://dx.doi.org/10.7717/peerj.11486#supplemental-information.

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
