# Peer review of "Biotin-streptavidin-guided two-step pretargeting approach using PLGA for molecular ultrasound imaging and chemotherapy for ovarian cancer"

_PeerJ, doi:10.7717/peerj.11486_

## Round 0.1 · original submission · Major Revisions

Dear Dr. Zhou,

Please change the paper according to the reviewers' comments.

thanks
best regards

Ferdinand

Reviewer 1 ·

Basic reporting

The authors developed Biotin-streptavidin-guided two-step pretargeting approach using PLGA for molecular ultrasound imaging and chemotherapy for ovarian cancer. The major concern needs to be addressed before the paper is accepted for publication. Followings are recommended for the manuscript.
What is the novelty of work?
Purpose of study in the abstract is not clear. Therefore, the author needs to a breakdown in two sentences and make it clear.
The objective of the study should be incorporated at the end of the introduction.
If authors have used reported HPLC method for drug analysis then it should be incorporated in the manuscript such as the composition of mobile phase and instrument specification.

The vocabularies, grammar and minor writing style in the manuscript must be revised and improved

Experimental design

Please indicate both the manufacturer’s name and location (including city, state, and country) for all specialized equipment, kits, software, incubators, instruments, and reagents used in the experiment wherever required.

Authors should provide the process parameters, e.g. how much volume of the organic phase, temperature, stirring speed, the power capacity of sonicator and so on.

How is the stability of formulations physiological condition? How is the biosafety of formulations?

Authors did not show any experiments in which they used blank nanoparticles. Nanoparticles itself could induce an unwanted cytotoxic effect on cells. MTT assay should be performed also with blank nanoparticles.

The author needs to justify why the cytotoxic effects of formulations on at least one normal cell line were not assessed.

Validity of the findings

No comment

Additional comments

Discussion of the results especially formulations is laconic, the author needs to improve the discussion with compare to previous literature.

The quality of the Figures is poor. Please provide high resolution (300 dpi) images.

Reviewer 2 ·

Basic reporting

no comment

Experimental design

no comment

Validity of the findings

no comment

Additional comments

This article is entitled " Biotin-streptavidin-guided two-step pretargeting approach using PLGA for molecular ultrasound imaging and chemotherapy for ovarian cancer " by Zhou, et al. They report a pre-targeting strategy using streptavidin (SA) and paclitaxel (PTX)-loaded phase-shifting poly lactic-co-glycolic acid (PLGA) nanoparticles with perfluoro-n-pentane (PTX-PLGA-SA/PFPs) on the treatment and ultrasound imaging of ovarian cancer. The concept of pre-targeting strategy using streptavidin (SA) was quite interesting. Overall, I find this design and the scope of this paper is suitable for the journal after a major revision.
1. There are no animal experiments to verify the results of in vitro experiments, I advise the author should supply the in vivo experiments to verify the effects of PTX-PLGA-SA/PFPs on the treatment and ultrasound imaging of ovarian cancer.
2. The author reported that PTX-PLGA/PFPs were prepared with a single emulsion (O/W) solvent evaporation method, as far as I know, the PLGA nanoparticles were usually prepared with a double emulsion(O/W/O), why you choose a single emulsion (O/W) solvent evaporation method? Please explain this.
3. The annotation of Figures cannot be seen clearly (Fig 1e, 1f, Fig 1g,1h, ect). The author should enlarge font.
4. The SEM and TEM in figure 1c,1d lacks bar, and some other figures.
5. Flow cytometry analysis of the connection efficiency of both SA and Ab bound to PTX-PLGA/PFPs in Figure, What is the sample size?
6. The annotation of CLSM pictures in Figure 4b is wrong, please revise.

---

## Round 0.2 · accepted · Accept

Dear Authors,

Thank you for a well-performed revision of the manuscript.

Thanks

Best regards

Ferdinand Frauscher

Reviewer 1 ·

Basic reporting

The authors have addressed all the concerns, improving the manuscript. In my opinion, the manuscript is now acceptable for publication

Experimental design

The authors have addressed all the concerns, improving the manuscript. In my opinion, the manuscript is now acceptable for publication

Validity of the findings

The authors have addressed all the concerns, improving the manuscript. In my opinion, the manuscript is now acceptable for publication

Additional comments

The authors have addressed all the concerns, improving the manuscript. In my opinion, the manuscript is now acceptable for publication